# Deep Convolutional Neural Networks Provide Motion Grading for High-Resolution Peripheral Quantitative Computed Tomography of the Scaphoid

**DOI:** 10.3390/diagnostics14050568

**Published:** 2024-03-06

**Authors:** Stefan Benedikt, Philipp Zelger, Lukas Horling, Kerstin Stock, Johannes Pallua, Michael Schirmer, Gerald Degenhart, Alexander Ruzicka, Rohit Arora

**Affiliations:** 1Department of Orthopedics and Traumatology, University Hospital Innsbruck, Anichstraße 35, 6020 Innsbruck, Austria; 2Department of Otorhinolaryngology, Hearing, Speech & Voice Disorders, University Hospital Innsbruck, Anichstraße 35, 6020 Innsbruck, Austria; 3Medical University of Innsbruck, Anichstraße 35, 6020 Innsbruck, Austria; 4Office Dr. Schirmer, 6060 Hall, Austria; 5Department of Radiology, University Hospital Innsbruck, Anichstraße 35, 6020 Innsbruck, Austria

**Keywords:** artifacts, artificial intelligence, convolutional neural networks, deep learning, image quality, motion grading

## Abstract

In vivo high-resolution peripheral quantitative computed tomography (HR-pQCT) studies on bone characteristics are limited, partly due to the lack of standardized and objective techniques to describe motion artifacts responsible for lower-quality images. This study investigates the ability of such deep-learning techniques to assess image quality in HR-pQCT datasets of human scaphoids. In total, 1451 stacks of 482 scaphoid images from 53 patients, each with up to six follow-ups within one year, and each with one non-displaced fractured and one contralateral intact scaphoid, were independently graded by three observers using a visual grading scale for motion artifacts. A 3D-CNN was used to assess image quality. The accuracy of the 3D-CNN to assess the image quality compared to the mean results of three skilled operators was between 92% and 96%. The 3D-CNN classifier reached an ROC-AUC score of 0.94. The average assessment time for one scaphoid was 2.5 s. This study demonstrates that a deep-learning approach for rating radiological image quality provides objective assessments of motion grading for the scaphoid with a high accuracy and a short assessment time. In the future, such a 3D-CNN approach can be used as a resource-saving and cost-effective tool to classify the image quality of HR-pQCT datasets in a reliable, reproducible and objective way.

## 1. Introduction

Scaphoid fractures represent the most common fractures of the carpus. Patients with untreated or missed scaphoid fractures risk developing non-unions, which may lead to severe wrist joint osteoarthritis with pain and functional deficits [1]. The risk of developing such scaphoid non-union is 2–5% [2]. Of note, 80% of these patients with scaphoid non-union receive an incorrect diagnosis [3]. Therefore, reliable diagnostic tools are essential for early diagnosis and evaluation of fracture healing during follow-up.

X-ray, computed tomography (CT), and magnetic resonance imaging (MRI) are established diagnostic imaging methods. MRI shows the highest sensitivity and specificity for fracture detection but is more expensive and less readily available. CT is often preferred, with high specificity but lower sensitivity than MRI [1,4,5].

In recent years, high-resolution peripheral quantitative computed tomography (HR-pQCT) has proven to be an innovative diagnostic tool for detecting fractures of the scaphoid [6,7,8], as well as for the evaluation of microarchitectural changes of the scaphoid during fracture healing [9]. It is a non-invasive method for in vivo three-dimensional (3D) imaging of distal extremity sections with the best signal-to-noise ratio and the highest spatial resolution of all tools used in in vivo diagnostics, with an in vivo voxel size of 61 µm. Radiosensitive organs are thereby only marginally exposed. The effective radiation dose for the patient is less than five µSv per stack [10,11,12].

Imaging artifacts are major limitations of HR-pQCT measurements that can hamper the precision and reproducibility of HR-pQCT measurements by patient movements [13,14,15,16,17,18], particularly at the scaphoid bone. This was already confirmed in an earlier study revealing a considerable influence of motion on bone morphometry parameters of the scaphoid [19]. A certain amount of movement occurs in every individual (e.g., coughing, breathing, resting tremor, nervousness). The manufacturer is well aware of this important issue and provides a visual grading scale (VGS), as described by Sode et al. [14], to assess the extent of motion artifacts that classifies the artifacts into five quality grades ranging from “grade 1” for no visible motion artifacts to “grade 5” for severe motion artifacts. It is essential to correctly classify the artifacts, as any visual motion artifact causes significant falsification of the quantitative parameters [8,18,19]. For time reasons, however, operators usually focus on grading only specific slices of the HR-pQCT scan rather than the entire scan. Moreover, these grading results are always observer dependent.

Therefore, this non-standardized and subjective approach leads to distinctive operator disagreement [17,18,20,21]. As a result, low-quality HR-pQCT scans might be considered of sufficient quality and vice versa; good-quality scans might be regarded as insufficient, providing inaccurate and incomplete imaging data sets. Overall, interobserver and intra-observer reliability in scaphoids is only fair to moderate. Poor image quality then influences the quantitative parameters of the scaphoid, with deviations of up to 20%. [19]

As a consequence, alternatives to the manual grading of CT images are urgently needed, and important efforts have already been made to develop alternatives to the manual grading of CT images [17,21,22,23,24]. All these studies combine the idea of using a data- or feature-driven approach to objectively grade image quality. Nowadays, neural networks, especially convolutional neural networks (CNNs), are helpful tools for rating and analyzing CT data [25]. Walle et al. [17] already used a CNN to grade HR-pQCT scans, but this approach analyzed the structure of single slices to grade the image quality. This may weaken the ability to detect artifacts in the axial scanning direction.

Since the artifacts and quality issues are expected to be present in three dimensions, this study assessed a three-dimensional (3D)-CNN to rate motion artifacts in HR-pQCT scans of the scaphoid.

The purpose of this study was to evaluate whether a machine-learning approach, specifically a three-dimensional convolution (3D-CNN) approach, is suitable for assessing the image quality of microCT data. This work investigates whether the results are consistent with a majority decision of three expert judgements. Our aim was to develop a system that allows a (1) quick, (2) precise and (3) examiner-independent evaluation of motion artifacts in scaphoid scans.

To achieve this aim, supervised (classification layer) and unsupervised deep-learning (autoencoder) methods were combined to achieve maximum precision. The core of the network is based on the use of three-dimensional convolutional kernels to account for variations in all dimensions.

## 2. Materials and Methods

### 2.1. Study Design and Population

This study is a retrospective data analysis of follow-up HR-pQCT scans from 53 patients, each with one non-displaced fractured and one intact scaphoid. The project was conducted in accordance with the Declaration of Helsinki (as revised in 2013) and was approved by the institutional ethics board of the Medical University of Innsbruck, Austria (No. 1259/2017). Informed and written consent was obtained from all patients. All patients were older than 18 years.

Figure 1 visualizes the six follow-ups with the number of assessed scans. During follow-up of one year, a total of six bilateral scans were planned per patient at 2, 4, 6, 12 weeks and 6 and 12 months after trauma. Cast immobilization on the fractured side ranged between four and twelve weeks after trauma. Scans with the fractured scaphoids were obtained with the wrist immobilized in a fiberglass cast at the 2-, 4- and sometimes at the 6-week follow-ups, which is known to have no considerable effects on interobserver variability [19].

Thirty-four patients were male, while nineteen were female. The median age was 28 (25% percentile: 24; 75% percentile: 48). A total of 482 scaphoid scans were evaluated, 242 from the fractured scaphoids and 240 from the non-fractured scaphoids. In total, 154 scaphoid scans were missing, as patients did not appear to each follow-up. From the 482 scaphoids, 1451 stacks were obtained for further analyses.

### 2.2. Scan Acquisition

All scans were performed using a second-generation HR-pQCT (XtremeCTII, Scanco Medical, Wangen-Brüttisellen, Switzerland). Three to four stacks of 10.2 mm (168 slices) were necessary to fully visualize the scaphoid. Using the anterior–posterior scout view, the scaphoid was centered within the stacks (Figure 2). To reduce patient movements during scanning, standard motion-restraining holders and the provided inflatable pads were used to immobilize the wrist in a thumb-up position [6,18,19]. Standard pre-settings were taken from the radius protocol provided by the manufacturer with a resolution of 60.7 µm isovoxels, an integration time of 46 ms, a current of 1460 µA and a voltage of 68 kV. The scan time for one stack was approximately 2 min at a radiation dose of 5 µSv. Daily monitoring by scanning a quality-control phantom was performed to ensure the longitudinal stability of the system.

### 2.3. Image Quality Grading

Using the Scanco Medical software package V6.1 provided by the manufacturer, the scans were cropped to the scaphoids and exported as AIM (advanced integrated matrix) files. The visual grading was performed with the image processing software ImageJ Version 1.49 (https://imagej.nih.gov; accessed on 1 October 2021, National Institutes of Health, United States of America). Three experienced examiners independently assessed all axial slices according to the visual grading scale described by Sode et al. [14], ranging from grade 1 (no visible artifacts) to grade 5 (severe artifacts) (Figure 3). The most severe motion artifact determined the quality grade of each stack. The final image quality of a scan was assessed using the median results of the three examiners.

### 2.4. Machine-Learning Approach

The categorization of HR-pQCT data was performed according to the visual grading scale described by Sode et al. [14], as shown in Figure 3. The HR-pQCT data are provided as three-dimensional voxel data. Artifacts are described by a convolution of the data by a Gaussian kernel. The nature of these movement artifacts and their possible mathematical description led to using a CNN for analyzing and categorizing movement artifacts or performing a more general quality assessment. Since the artifacts and quality issues are expected to be present in three dimensions, the choice for the basic structure of the neural network was made for a CNN with three-dimensional kernels (3D-CNN).

The network architecture was based on a ResNet architecture with several convolutional layers intercepting by pooling layers and then fully connected layers intercepted by dropout layers (as schematically shown in Figure 4). The network was finished by five sigmoid layers corresponding to the five distinct quality classes. Thus, the 3D-CNN used consisted of four (25 × 25 × 5 × 16; 20 × 20 × 3 × 32; 10 × 10 × 2 × 64; 5 × 5 × 1 × 125) convolutional layers and two 3D MaxPooling layers. After a flattening operation, this was followed by three fully connected layers with 50, 25, and 5 neurons intercepted by dropout layers. The dropout rate was 12.5%. The final layer consisted of five output neurons with a sigmoid activation function. The network consisted of ~100,000 parameters.

The neural network benefits from pre-training on a larger dataset (concept shown in Figure 5). This training dataset consists of the HR-pQCT images of the best and second-best quality classes, with a noise component added by convolution with three-dimensional Gaussian kernels. This procedure introduces smearing, similar to the artifacts caused by the patient’s movements, and can therefore be used to pre-train the neural network. This artificial image quality degradation is randomly added at five distortion levels that mimic the quality levels of the real world data. The three-dimensional orientation of the Gaussian kernel was also randomly altered, which introduces artifacts that are randomly oriented in all three dimensions. This artificial dataset enables the 3D-CNN to learn the features of randomly smeared images.

The next step is to refine the pre-trained neural network on the actual dataset. This procedure is referred to as transfer learning. The weights or pre-trained parameters of the network with the artificially treated data are used as a starting point for the training of the final neural network. The pre-trained neural network is refined using the actual patient data and it is trained to classify the quality categories of the data. The dataset was split into training and test datasets following an 80/20 percent split. The workflow of the experiment is shown in Figure 5.

In order to gain insight into the internal operations of the CNN, the outputs of the convolutional layers for a given input dataset were further examined. By examining the output of the first layer, some understanding of the neural network can be obtained. The required time for the rating of one scaphoid by the 3D-CNN was determined in seconds.

## 3. Results

### 3.1. Confusion Matrix with High Accuracy

The confusion matrix of the deep-learning-based classification is shown in Figure 6. These results show a high degree of agreement of 92% and 95% (bright diagonal entries) with the result of the image quality grading by the three trained operators. The 3D-CNN classifier reached a ROC-AUC score of 0.94 as well as specificity and precision values of 0.93 and 0.91, respectively. There was only some cross-talk: the first class, for instance, showed 3% wrong categorizations in the second and 2% in the third predicted class.

### 3.2. Suggested Focus of the Neural Network

As the network grows deeper, the outcomes of the convolutional filters become increasingly abstract, making them difficult to explain using descriptive measures. Figure 7A displays the section of a CT dataset as an illustrative example. Figure 7B–F depict the response of selected convolutional kernels from the initial layer. The response of these kernels to the input data suggests that the neural network primarily focuses on detecting edges and boundaries.

### 3.3. Duration of the 3D-CNN Procedure

As mentioned above, the final layer consists of five output neurons with a sigmoid activation function. The network consists of ~100,000 parameters. The required time required for the rating of one single scaphoid dataset was an average of 2.5 s.

## 4. Discussion

This study demonstrates the ability of a 3D neural network to rapidly assess and accurately quantify patient motion in HR-pQCT scans compared to standard manual rating procedures. The accuracy of the 3D-CNN to predict the image quality rating was 92% to 95% with an assessment time of only 2.5 s per scaphoid. The importance of such an automated rating system becomes apparent when considering the variability and bias of operator-based motion scoring systems [17,18,20,21]. As the abilities of a neural network strongly depend on the quantity of the training data and the quality of the training labels, we collected a large set of training datasets (1451 stacks), all graded by three experienced and independent professionals. This prevents the influence of bias by only one operator and leads to a more reliable data set and target vector. Moreover, not only certain slides but every single slide was assessed by the three observers, which led to an even more precise grading of the source material with more reliable results.

Especially in the scaphoid, motion artifacts are a severe problem, as they can lead to a significant bias among the quantitative measurements [19]. This can be explained by the anatomy of the scaphoid, which is different compared to the distal radius or the tibia. From the macroscopic aspect, the scaphoid is much smaller and more complex. Regarding the quantitative parameters, scaphoids seem to have a partly thinner cortical shell and a higher degree of mineralization compared to other bones [6,26,27,28]. Especially the cortical shell may cause problems in visual grading, as interruptions in the cortex might be detected earlier, as in larger bones with thicker cortical substance. This might also explain the response of the kernels to the input data, which suggests that the neural network primarily focuses on detecting edges and boundaries. Those tend to appear elongated in CT slices with lower quality.

Significant subjectivity in operator-based grading and the negative impact of poor-quality images on the quantitative data was frequently described: interobserver and intra-observer variability were only fair to moderate, with kappa values ranging between 0.37 and 0.47 in analyses of 759 scaphoid stacks from 22 patients compared to the results from three independent observers [19]. Images with grade 5 ratings had a significantly different outcome regarding the quantitative parameters than grade 1 images (deviations up to ~22%). In another study comparing the scaphoid’s standard and post hoc grading, the standard grading missed 85.7% of poor-quality scans [6]. However, since the standard grading only uses single low-resolution preview slices, these data are only to a limited extend comparable to the other studies. With kappa values of 0.57 for the interobserver and 0.68 and 0.74 for the intra-observer reliability, agreement was higher in studies focusing on the distal radius and the distal tibia [18]. Also, the examination of the distal radii and distal tibiae resulted in an interclass correlation of 0.77 between four graders of the same laboratory and two graders from two external laboratories, indicating good rater agreement [21].

Without doubt, scans with the best possible image quality should form the basis for every HR-pQCT study. Rapid grading of motion is desirable, as repeated scanning becomes necessary in case of poor image quality due to motion artifacts [18,21]. With only 2.5 s per scaphoid dataset, the CNN was much faster than any human operator. Manual grading of a scaphoid scan with over 300 slices can take several minutes per examiner if every single slice is evaluated individually. This fact can be considered an important advantage of a CNN applied in clinical practice to reduce assessment times, although other standards like image acquisition using the manufacturer’s standard motion-restraining holders with the appropriate inflatable pads should be applied, too [6,18,19], with or without cast immobilization. A fiberglass cast, as used in our study, generally negatively impacts image quality less than a plaster-of-Paris cast [29]. Overall, a standardized and reproducible study protocol and an experienced team are obligatory.

The approach discussed by Walle et al. [17] showed similar, although slightly lower, categorization abilities. They followed an ensemble based deep neural network approach based on 2D convolutions (2D CNN) for the analysis of motion artifacts in distal radius HR-pQCT scans (second generation). A total of 90 participants’ healthy distal radius images from a previous distal radius fracture database were included. The intact collateral radius was scanned 1,3, 5, 13, 26 and 52 weeks after trauma. Median age was 56, and the male-to-female ratio was 1:2. Manual visual grading was performed by two graders using the same visual grading classification as in the current study. The datasets were divided by patient into 60% training, 20% validation, and 20% test datasets by randomization. The authors reported a precision of ~91% and a recall of ~89%. They discussed a significant uncertainty for the third of five image quality classes. The approach presented in this paper did not show any such uncertainty focused on a single quality class. A possible reason for this might be found in the use of three-dimensional convolutions, which might add stability to the network.

The mathematical structure of the convolutional operations used in the neural network suggests that such a neural network may also be suitable for analyzing other images, such as MRI. In MRIs, a 90% accuracy in the detection of motion artifacts was shown when combining CNN with manual analyses by an additional operator, further improving the CNN-based grading [17,30]. Others reported a classification accuracy of 88.3–93.8% in MRIs [17,31]. Their CNNs more accurately classified severe motion artifacts than smaller ones. The reason might be that small artifacts are harder to distinguish based on their appearance concerning the pixel or voxel size, which, in turn, might depend on the hardware used. These deep-learning-based approaches are based on analysis of the raw data. An alternative approach was, for instance, to use decision tree classifiers that were trained on features extracted from the CT datasets [22]. Using features and decision trees allows for a more straightforward and more readable explanation of the algorithm’s inner workings.

The main limitation of the 3D-CNN approach is the cross-talk between neighboring image quality classes, as shown by the non-diagonal elements in Figure 6. This cross-talk is most likely due to the discrete categorization of a continuous spectrum of image quality issues. Mapping this continuous spectrum onto a discrete class-based spectrum may introduce those issues. The same problems, i.e., difficulties in deciding on a quality class for images between two classes, are also present for human operators. Extending the machine-learning approach to a continuous quality rating might help to solve such a problem. In clinical practice, however, such a score would have to be transformed back into a categorical variable, again deleting its benefit. Other restrictions of this approach are that the data used are all from the same clinical environment, i.e., the same devices were used, only scaphoid bones were examined, etc. Finally, only individuals aged over 18 were examined. During early development, carpal bones have a different anatomy [32], which could influence the visual grading. An expansion of the training dataset to other bones and, in the best case, other clinics may increase the neural network’s abilities to capture artifacts and deal with sample variance. Such a step would furthermore allow for a more sophisticated analysis of the neural network’s weaknesses and properties.

## 5. Conclusions

This study proposes a reliable deep-learning approach for rating radiological images with high accuracy and short assessment times. This tool can be used to reduce the amount of work and time required for this process and can therefore be considered a resource-saving and cost-effective option. Furthermore, the results are objective and reproducible and are not influenced by the examiner’s experience or by technical influences such as screen resolution and brightness. An extension of this study with an extended training set will allow for further improvements in the 3D neural network as a reliable general image quality rating tool.

## Figures and Tables

**Figure 1 diagnostics-14-00568-f001:**
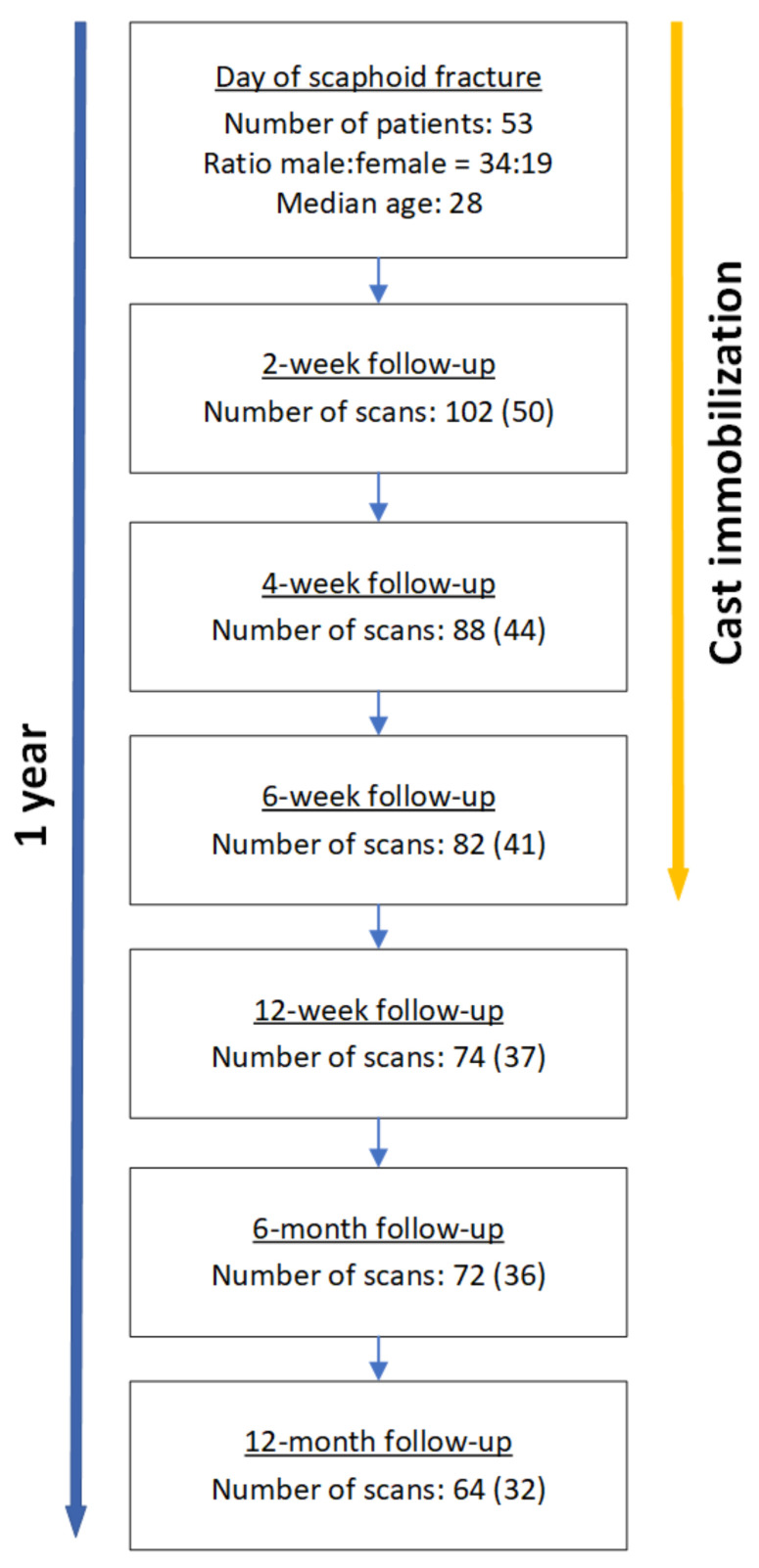
Flow chart visualizing the six follow-ups with the number of assessed scans. Both wrists were scanned if possible. The fractured side was scanned in a fiberglass cast during the first three follow-ups. The number of scans of the healthy wrist is given in brackets.

**Figure 2 diagnostics-14-00568-f002:**
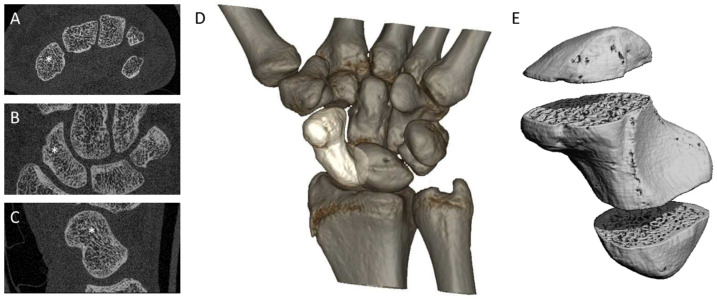
Region of interest. (**A**–**C**) axial, frontal and sagittal views of the scaphoid marked with a white asterisk; (**D**) localization of the scaphoid in the carpus; (**E**) 3D view of the scaphoid divided into its three stacks.

**Figure 3 diagnostics-14-00568-f003:**
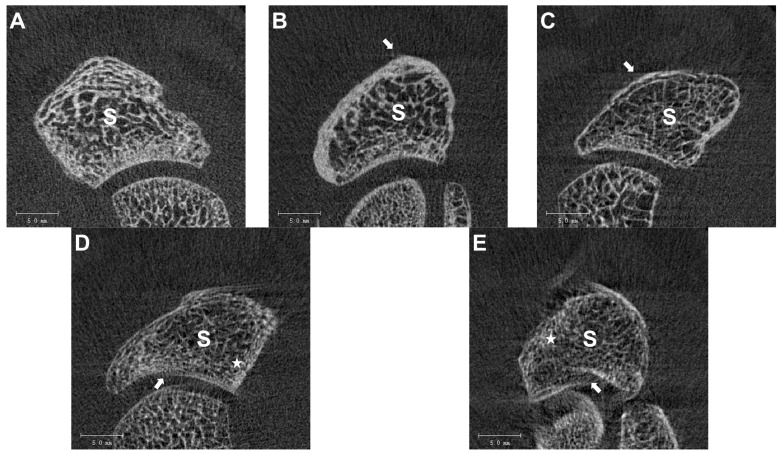
Visual grading scale of the scaphoid: (**A**) Grade 1, no visible motion artifacts. (**B**) Grade 2, slight horizontal streaks (white arrow). (**C**) Grade 3, prominent horizontal streaks (white arrow), intact cortex. (**D**) Grade 4, prominent horizontal streaks, minor disruptions of the cortex continuity (white arrow), and minor trabeculae smearing (white asterisk). (**E**) Grade 5, prominent horizontal streaks, major disruption of the cortical continuity (white arrow), major trabecular smearing (white asterisk). S, scaphoid.

**Figure 4 diagnostics-14-00568-f004:**
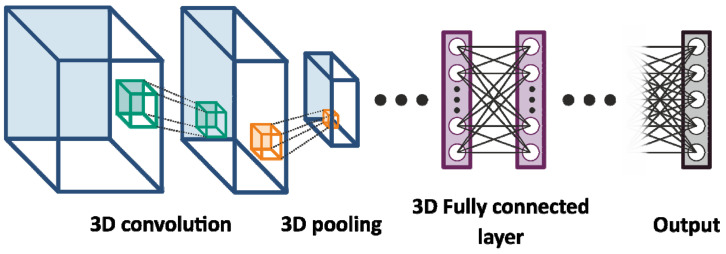
Schematic representation of the neural network. The neural network consists of four convolutional and two 3D MaxPooling layers, followed by three fully connected layers intersecting by dropout layers. The final layer consists of five output neurons with a sigmoid activation function.

**Figure 5 diagnostics-14-00568-f005:**
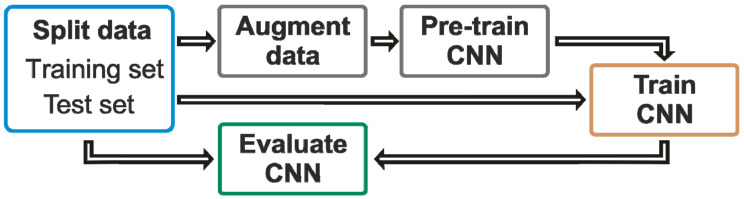
Schematic representation of the training process. In the first step, the dataset is split into a training and a test set (blue). The test set is then augmented (random rotations in 90° steps and mirroring over all three dimensions) and used to pre-train the neural network (gray). This pre-trained neural network is then further trained using the original dataset to classify the quality categories of the data (brown). The performance of the neural network is analyzed using the test set (green).

**Figure 6 diagnostics-14-00568-f006:**
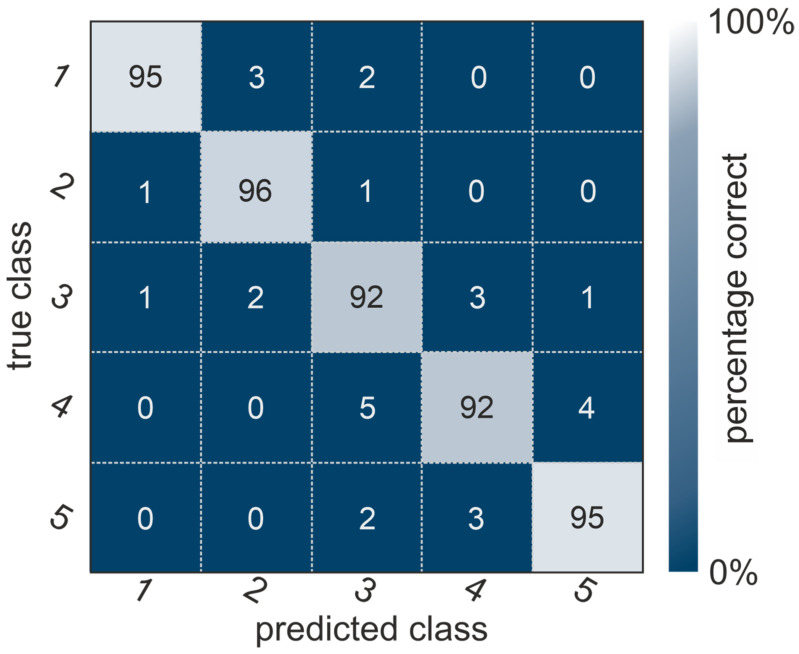
Confusion matrix of the quality class classification. The correct class is predicted for more than 92% of all datasets. Some cross-talk occurs between the quality classes (with sums up to 5%).

**Figure 7 diagnostics-14-00568-f007:**
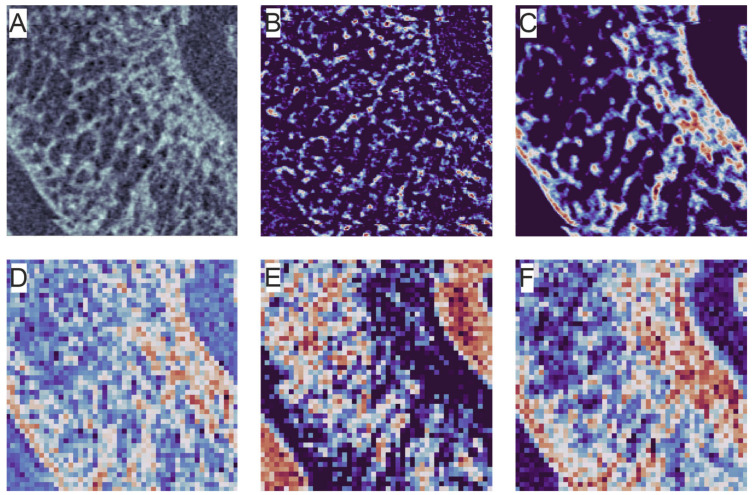
Illustrative example of a section of a CT dataset. (**A**) Example of a microCT slice. (**B**–**F**) The images (**B**–**F**) show the response of the first layer of the neural network to the input image shown in (**A**).

## Data Availability

The data presented in this study are available on request from the corresponding author. The data are not publicly available due to scientific reasons.

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
