# Peer review of "Deep Convolutional Neural Networks Provide Motion Grading for High-Resolution Peripheral Quantitative Computed Tomography of the Scaphoid"

_diagnostics, 2024, doi:10.3390/diagnostics14050568_

Round 1

Reviewer 1 Report

Comments and Suggestions for Authors

In this article, the authors present a robust deep-learning methodology designed for the accurate evaluation of radiological images within a brief timeframe.

Overall, the article presents a promising and potentially impactful contribution to the field, and addressing the following improvements would further strengthen its quality and significance:

-          Clearly highlight the main contributions of this work in the introduction section using bullet points.

-          Add the paper's organization details at the end of the introduction.

-          Improve the organization of the manuscript (e.g., justify text).

-          Extend the conclusion section by providing additional details.

-          Conduct an ablation analysis to evaluate the performance of the suggested approach.

-          Include a comparison with other related works.

-          Define how ground truth images are determined.

-          Consider using a well-known and large dataset.

Reviewer 2 Report

Comments and Suggestions for Authors

This manuscript classifies the worsening artifact occurring in the scaphoid into 5 grades using CNN. The classification accuracy has been evaluated using a confusion matrix.

1. What is the disease causing this scaphoid artifact? The authors mention that the worsening artifact could lead to Osteoarthritis development (in hand?). On the other hand, the subjects involved in this study are older than 18 years old.

2. The reference below may enrich the discussion related to scaphoid or carpal bone.

A Faisal, A Khalil, HY Chai, KW Lai, X-ray carpal bone segmentation and area measurement. Multimedia Tools and Applications 81 (26), 37321–37332

2. The authors should also present the precision and specificity numbers.

3. The authors should also compare the classification accuracy obtained by other works.

4. What is the implication obtained from this study? For example, may it reduce the workload of the physician in analyzing big data? Please add the implications to the abstract and conclusion.

Comments on the Quality of English Language

-

Reviewer 3 Report

Comments and Suggestions for Authors

The study focused on deep-learning-based classification of radiological images for finer assessments. Authors have used 3D CNN architecture to build the model with inputs such as scaphoid scan images. The method section describes the workflow nicely.  The introduction is well-written, properly discusses the research gap, and builds the hypothesis for the current study. I have the following comments on the study-

1) Do you have the gender information of all the individuals considered in the study? Do you think that gender biases are ignorable in this study? If so, it is just fine to provide gender information and reasons for ignoring their biases in the "material and method" section.

2) In the "study design and population," is it possible to make a flow chart diagram explaining the samples, follow-ups, days, and no if total scaphoid scans that were considered for further analysis? It is easy to visualize this information.

3) In Figure 4, less explanation was provided on data augmentation.

4) Figure 5, put like 92 instead 0.92, because the color code represents %.

Comments on the Quality of English Language

Minor editing is required

Round 2

Reviewer 1 Report

Comments and Suggestions for Authors

The manuscript can be accepted in the current form.

Reviewer 3 Report

Comments and Suggestions for Authors

Thanks for addressing all the comments. 

Just a suggestion. In the cover letter, it is better to provide the line number or paragraph where changes were made to track them easily. 

Comments on the Quality of English Language

Minor editing is required.